# `InstOptima`: Evolutionary Multi-objective Instruction Optimization via Large Language Model-based Instruction Operators

**Heng Yang[1], Ke Li[1]**

[1]Department of Computer Science, University of Exeter, EX4 4QF, Exeter, UK

`{hy345, k.li}@exeter.ac.uk`

## Abstract

Instruction-based language modeling has received significant attention in pretrained language models. However, the efficiency of instruction engineering remains low and hinders the development of instruction studies. Recent studies have focused on automating instruction generation, but they primarily aim to improve performance without considering other crucial objectives that impact instruction quality, such as instruction length and perplexity. Therefore, we propose a novel approach (i.e., `InstOptima`) that treats instruction generation as an evolutionary multi-objective optimization problem. In contrast to text edition-based methods, our approach utilizes a large language model (LLM) to simulate instruction operators, including mutation and crossover. Furthermore, we introduce an objective-guided mechanism for these operators, allowing the LLM to comprehend the objectives and enhance the quality of the generated instructions. Experimental results demonstrate improved fine-tuning performance and the generation of a diverse set of high-quality instructions.

## 1 Introduction

With the rapid development of language models (Ouyang et al., 2022; Touvron et al., 2023; OpenAI, 2023), instructions (also known as prompts) play a crucial role in instruction-based language modeling, and different instructions may lead to significant differences in model outputs (Zhou et al., 2022; Honovich et al., 2022; Wan et al., 2023). For instance, even slightly perturbed instructions (e.g., synonym substitutions (Wang et al., 2021; Zhou et al., 2021) or adversarial attacks (Wan et al., 2023; Zhu et al., 2023)) can result in unexpectedly low performance. However, there are three problems regarding instruction-based learning that still need to be addressed in existing works.

Firstly, existing works (Lester et al., 2021; Gu et al., 2022; Zhou et al., 2022, 2023; Li et al.,

2023; Chen et al., 2023) aim to obtain a large number of instructions through automated instruction generation to filter high-performance instructions. However, due to the large and non-differentiable textual search space (Ishibashi et al., 2023; Cho et al., 2023), the automated instruction generation and instruction engineering methods (Brown et al., 2020; Liu et al., 2023) are inefficient and struggle to search for various high-quality instructions. Secondly, the objectives of instruction generation are not clear. Current research (Lester et al., 2021; Gu et al., 2022; Pitis et al., 2023) regards performance (i.e., metrics) as the sole criterion for instruction quality. However, model performance alone cannot precisely explain instruction quality. We propose to refine instruction quality by considering fine-grained objectives, such as length and perplexity. Shorter instructions can lower computational costs, especially for large-scale models and datasets. Lower perplexity indicates that instructions are more easily understood by language models. Lastly, the diversity of instructions has been neglected in existing studies, while increasing the diversity of instructions can mitigate adversarial attacks (Wan et al., 2023; Zhu et al., 2023) and improve instruction robustness (Yu et al., 2022; Zhu et al., 2023). We aim to obtain multiple alternative instructions based on multi-objective optimization, which can facilitate comprehensive evaluation of instructions.

To address these three problems, we formulate the task as an evolutionary multi-objective optimization problem and propose our framework called `InstOptima`. We leverage a large language model, specifically `ChatGPT` (OpenAI, 2023), to facilitate instruction operations such as mutation and crossover. Furthermore, we introduce an objective-guided mechanism to assist the language model in generating high-quality instructions. In terms of optimization objectives for instruction generation, `InstOptima` incorporates

three objectives: performance (metrics), length, and perplexity, enabling the exploration of a diverse and high-quality set of instructions. We adopt `NSGA-II` (Deb et al., 2002) in `InstOptima` to obtain a Pareto front of instruction sets.

To validate the efficacy of `InstOptima`, we conducted experiments on three generation-based classification tasks. The experimental results indicate that `InstOptima` can concurrently obtain a diverse set of instructions that outperform the counterparts regarding performance.

In summary, our contributions are as follows:

- We simulate instruction operators based on an LLM. We also show that the objective-guided operators help the LLM understand optimization objective values and improve instruction quality.
- We divide the orientation of instruction search into multiple objectives, such as performance, length, and perplexity, facilitating fine-grained control over instruction quality.
- We utilize a multi-objective optimization algorithm to automatically search for a set of high-quality instructions, which could benefit defending against adversarial attacks and improving instruction robustness.

The codes are available at: https://github.com/yangheng95/InstOptima.

## 2 Proposed Method

In this section, we first introduce the instruction-based text generation, followed by the details of `InstOptima`.

### 2.1 Instruction-based Generation

In text generation-based tasks[1], instructions are utilized to facilitate in-context learning (Brown et al., 2020) and improve language modeling. An instruction (depicted in the right part of Fig. 1) is represented as $\mathbf{I} = \text{Concat}(\mathbf{d}, \mathbf{e})$, where $\mathbf{d}$ and $\mathbf{e}$ are the definition and example of the target task, respectively. $\mathbf{d}$ and $\mathbf{e}$ are token sequences similar to $(\mathbf{x}, \mathbf{y}) \sim \mathcal{D}$, where $\mathbf{x}$, $\mathbf{y}$, and $\mathcal{D}$ denote the input, output, and task dataset, respectively. The modeling of a generation model $f(\cdot, \cdot)$ is defined as follows:

$$\hat{\mathbf{y}} = f(\mathbf{x}, \mathbf{I}) \qquad (1)$$

where $\hat{\mathbf{y}}$ represents the generated output given $\mathbf{x}$ and $\mathbf{I}$. In `InstOptima`, we aim to address the problem of automated instruction generation through multi-objective optimization.

### 2.2 Evolutionary Instruction Optimization

The workflow of `InstOptima` is illustrated in Fig. 1. We begin by initializing a parent population of instructions to start evolving. The parent population is manipulated by LLM-based operators to generate offspring. Subsequently, we employ the non-dominated sort algorithm to rank the combined population and measure the crowdness of instructions. At the end of each generation, we randomly replace some Pareto-front instructions with new instructions to enhance the diversity of the population (referred to as genes in `NSGA-II`). We also provide the pseudo code of the `InstOptima` in Appendix A.4.

#### 2.2.1 Operators for Instructions

To handle the non-differentiable text search space, we formulate these operators as a text generation task based on `ChatGPT`. In other words, we define a set of fixed prompts $\tilde{\mathbf{P}}$, $\tilde{\mathbf{P}} = \{\tilde{P}_{dm}, \tilde{P}_{dc}, \tilde{P}_{em}, \tilde{P}_{ec}\}$, to guide `ChatGPT` in performing the instructions, where $\tilde{P}_{dm}, \tilde{P}_{dc}, \tilde{P}_{em}, \tilde{P}_{ec}$ are the fixed prompts for the four operations:

- **Definition Mutation** ($\tilde{P}_{dm}$): This operator mutates the definition in an instruction. It can involve paraphrases and substitution of new definitions.
- **Definition Crossover** ($\tilde{P}_{dc}$): This operator combines the definitions of two instructions to create a new instruction. It can involve merging or exchanging parts of the definitions between the parent instructions.
- **Example Mutation** ($\tilde{P}_{em}$): This operator perturbs the example to introduce diversity. It can involve modifications such as example substitution, addition, or deletion.
- **Example Crossover** ($\tilde{P}_{ec}$): This operator randomly selects examples from two instructions to create a new instruction.

For instance, we formulate the mutation operation as follows:

$$\hat{\mathbf{d}}_{dm} = \text{ChatGPT}(\text{Concat}(\tilde{P}_{dm}, \mathbf{d})) \qquad (2)$$

where $\hat{\mathbf{d}}_{dm}$ is the new definition generated based on the original instruction $\mathbf{I}$. The new instruction is denoted as $\hat{\mathbf{I}}$, $\hat{\mathbf{I}} = \text{Concat}(\hat{\mathbf{d}}_{dm}, \mathbf{e})$. The other operators follow a similar formulation to mutation. Further details of the fixed prompts are available in Appendix A.5.

---

[1] We validate `InstOptima` generation-based text classification, and `InstOptima` can be easily applied to other instruction-based modeling tasks.

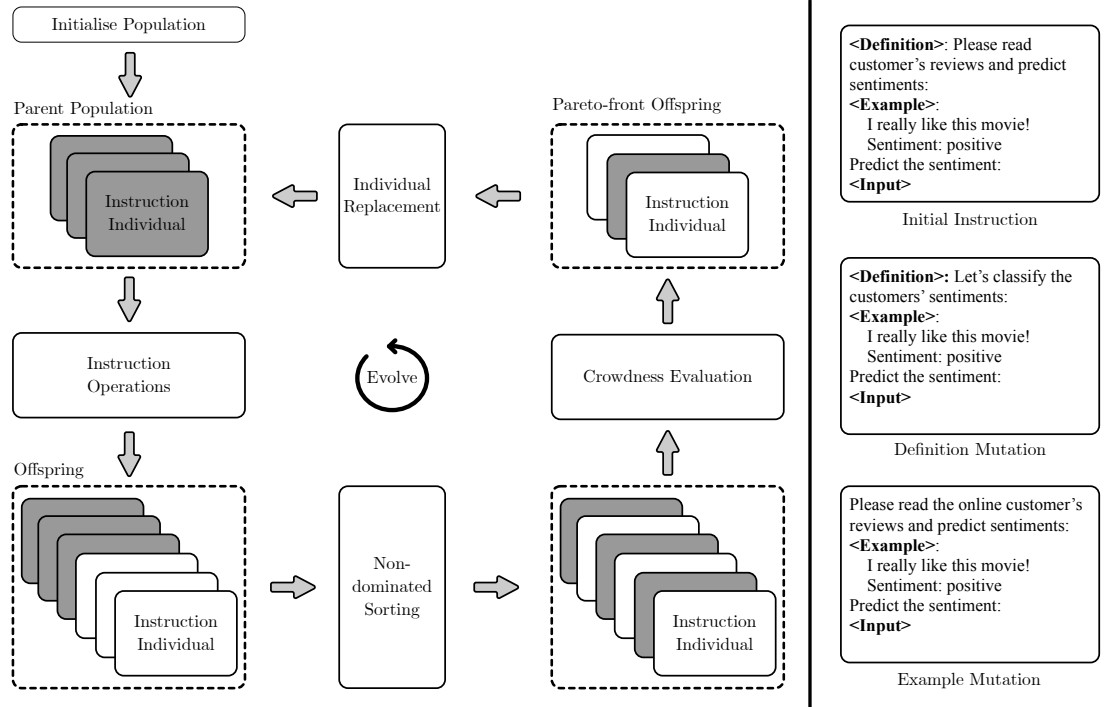

Figure 1: The main framework of `InstOptima` (left) and instruction operation examples (right). The details of the workflow that is explained in Section 2.2. The population is composed of individuals of instruction examples.

### 2.2.2 Optimization Objectives

We consider three objectives $\mathcal{F} = (m, l, r)$, in optimization, i.e., metrics ($m$), length ($l$), and perplexity ($r$) of the instruction.

- **Performance**: We use a set of metrics, such as accuracy, $f1$ score, precision, and recall, obtained by evaluating the instruction to calculate the performance objective. The performance objective is represented as the reciprocal of the sum of these metrics.
- **Length**: The length of the instruction is measured in terms of the number of characters. This measurement is fair regardless of the tokenization strategy.
- **Perplexity**: The perplexity of the instruction is measured using the RoBERTa model.

The evaluation of objectives $\mathcal{F}$ is shown in the pseudo-code in Appendix A.4 but not depicted in Fig. 1 for simplicity.

### 2.3 Objective-Guided Instruction Operators

To enhance the performance of `ChatGPT` through in-context learning, we propose a simple yet effective objective-feedback mechanism. Specifically, we incorporate the fitness values $\mathcal{F} = (m, l, r)$ into the fixed prompts. For example, we can append "*Please refer to the objective values:* $(\mathbf{d}_1, \mathcal{F}_1)$, $(\mathbf{d}_2, \mathcal{F}_2)$" to $\tilde{P}_{dc}$ in instruction examples crossover.

These operators[2] allow `ChatGPT` to autonomously decide to emphasize or down-weight an instruction based on the current objectives $\mathcal{F}$.

## 3 Experimental Setup

We conducted a comprehensive set of experiments[3] to validate the performance of `InstOptima`. The detailed experiments setups and implementations are described in Appendix A.1.

### 3.1 Baseline Methods

We used random instruction (`RanInstruct`) generation (i.e., request `ChatGPT` generates several instructions similar to instructions generated by `InstOptima`) and no-instruction (`NoInstruct`) as comparison baselines. The `RanInstruct` generates five random instructions using the LLM to evaluate the same three objectives as `InstOptima`. The `NoInstruct` ablates instruction in the classification-oriented fine-tuning of `Flan-T5`.

---

[2] Please refer to Table 4 for the actual implementations of these objective-guided operators.

[3] To improve the reproducibility, we release all experimental materials in the supplementary files of the submission, including source code, experiment logs, and results, optimized instructions.

Table 1: The experimental performance of `InstOptima`. We show the ACCURACY instead of the performance objective for intuitive evaluation. The symbols '↗' and '↘' indicate 'larger is better' and 'lower is better', respectively. We repeat each experiment in five rounds and report the average results. The best results are in **bold**. The ACCURACY is the best accuracy in the Pareto-front, while the LENGTH and PERPLEXITY are correlated with the instruction that achieves the best accuracy.

| MODEL | DATASET | InstOptima | | | RanInstruct | | | NoInstruct |
| --- | --- | --- | --- | --- | --- | --- | --- | --- |
| | | ACCURACY ↗ | LENGTH ↘ | PERPLEXITY ↘ | ACCURACY ↗ | LENGTH ↘ | PERPLEXITY ↘ | ACCURACY ↗ |
| FlanT5-small | Laptop14 | **84.9**$_{\pm0.2}$ | **622.6**$_{\pm51.5}$ | **1.07**$_{\pm0.02}$ | 82.5$_{\pm0.3}$ | 740.2$_{\pm84.6}$ | **1.07**$_{\pm0.05}$ | 53.8$_{\pm0.3}$ |
| | Restaurant14 | **84.9**$_{\pm0.2}$ | 421.6$_{\pm82.4}$ | **1.11**$_{\pm0.01}$ | 82.3$_{\pm0.4}$ | **328.5**$_{\pm38.5}$ | 1.15$_{\pm0.03}$ | 19.2$_{\pm0.4}$ |
| | SST2 | **89.7**$_{\pm0.1}$ | **402.7**$_{\pm39.1}$ | **1.09**$_{\pm0.01}$ | 88.7$_{\pm0.5}$ | 499.7$_{\pm73.2}$ | 1.16$_{\pm0.02}$ | 86.9$_{\pm0.1}$ |
| | AGNews | **90.2**$_{\pm0.1}$ | **452.5**$_{\pm27.7}$ | **1.11**$_{\pm0.04}$ | 82.9$_{\pm0.6}$ | 560.6$_{\pm28.7}$ | 1.12$_{\pm0.04}$ | 74.3$_{\pm0.1}$ |
| | SNLI | **69.1**$_{\pm0.2}$ | **295.3**$_{\pm74.8}$ | 1.14$_{\pm0.02}$ | 50.8$_{\pm0.5}$ | 507.3$_{\pm98.0}$ | **1.09**$_{\pm0.07}$ | 37.9$_{\pm0.2}$ |
| | MNLI | **57.4**$_{\pm0.3}$ | **385.8**$_{\pm57.5}$ | 1.12$_{\pm0.03}$ | 40.6$_{\pm1.1}$ | 519.7$_{\pm68.6}$ | **1.09**$_{\pm0.05}$ | 37.3$_{\pm0.3}$ |
| FlanT5-base | Laptop14 | **88.4**$_{\pm0.3}$ | **207.2**$_{\pm57.3}$ | **1.04**$_{\pm0.04}$ | 86.6$_{\pm0.3}$ | 549.7$_{\pm85.7}$ | 1.10$_{\pm0.03}$ | 62.3$_{\pm0.2}$ |
| | Restaurant14 | **89.1**$_{\pm0.2}$ | **359.4**$_{\pm39.7}$ | **1.06**$_{\pm0.03}$ | 87.4$_{\pm0.5}$ | 589.3$_{\pm63.2}$ | 1.11$_{\pm0.03}$ | 52.8$_{\pm0.2}$ |
| | SST2 | **94.5**$_{\pm0.1}$ | 397.8$_{\pm69.4}$ | **1.08**$_{\pm0.01}$ | 93.0$_{\pm0.4}$ | **385.6**$_{\pm55.0}$ | 1.12$_{\pm0.01}$ | 92.6$_{\pm0.1}$ |
| | AGNews | **93.5**$_{\pm0.3}$ | **300.1**$_{\pm73.8}$ | **1.15**$_{\pm0.01}$ | 90.1$_{\pm0.6}$ | 485.4$_{\pm68.2}$ | 1.16$_{\pm0.02}$ | 88.1$_{\pm0.1}$ |
| | SNLI | **86.6**$_{\pm0.3}$ | 430.9$_{\pm82.2}$ | **1.10**$_{\pm0.02}$ | 86.4$_{\pm0.5}$ | **399.3**$_{\pm23.8}$ | 1.11$_{\pm0.04}$ | 85.9$_{\pm0.4}$ |
| | MNLI | **80.2**$_{\pm0.4}$ | 388.2$_{\pm58.8}$ | **1.11**$_{\pm0.03}$ | 77.8$_{\pm0.7}$ | 449.1$_{\pm70.3}$ | 1.20$_{\pm0.03}$ | 74.5$_{\pm0.4}$ |
| ChatGPT | Laptop14 | **83.2**$_{\pm2.2}$ | **512.9**$_{\pm51.5}$ | 1.08$_{\pm0.02}$ | 83.1$_{\pm0.8}$ | 877.6$_{\pm51.5}$ | **1.05**$_{\pm0.03}$ | 67.8$_{\pm5.8}$ |
| | Restaurant14 | **96.3**$_{\pm1.9}$ | **487.3**$_{\pm55.9}$ | 1.09$_{\pm0.02}$ | 92.1$_{\pm1.3}$ | 421.6$_{\pm82.4}$ | 1.10$_{\pm0.02}$ | 75.2$_{\pm6.1}$ |

## 3.2 Main Results

The results in Table 1 show the performance of `InstOptima`. Overall, `InstOptima` achieves superior objectives based on various base models (e.g., `ChatGPT` and `FlanT5`). For example, it outperforms all baselines on all datasets in terms of ACCURACY. However, for instruction LENGTH and PERPLEXITY, the `RanInstruct` sometimes achieves better objective values. On the other hand, `NoInstruct` performs poorly on all datasets in terms of ACCURACY, underscoring the importance of instructions in generation-based fine-tuning. Moreover, the ACCURACY objective exhibits small intervals but relatively large variances, making it more challenging to optimize. However, existing methods that prioritize performance optimization struggle to handle the variances in metrics. On the other hand, the LENGTH objective is easier to optimize due to its significant variations and greater significance. This is because long instructions can result in up to twice training times than short instructions. The PERPLEXITY metric ranges within small intervals, indicating a moderate optimization challenge, but it significantly impacts the understanding of instruction engineers. In addition to these three objectives, `InstOptima` can easily accommodate additional objectives for precise control of instruction generation.

Overall, `InstOptima` demonstrates impressive performance in instruction optimization across various tasks and datasets.

## 3.3 Research Questions

We further discuss our observations and analysis by answering several research questions.

**RQ1: Do the objective-guided operators help instruction optimization?**

Table 2: The experimental performance of `InstOptima-N` on `FlanT5-small`. The tokens "−" and "+" indicate worse and better objectives than `InstOptima`.

| DATASET | InstOptima-N | | |
| --- | --- | --- | --- |
| | ACCURACY ↗ | LENGTH ↘ | PERPLEXITY ↘ |
| Laptop14 | 84.4$_{\pm0.2}$ − | 789.3$_{\pm86.2}$ − | 1.07$_{\pm0.02}$ |
| Restaurant14 | 83.7$_{\pm0.3}$ − | 455.8$_{\pm79.9}$ − | 1.12$_{\pm0.03}$ − |
| SST2 | 89.6$_{\pm0.1}$ − | 435.2$_{\pm52.1}$ − | 1.12$_{\pm0.02}$ − |
| AGNews | 86.7$_{\pm0.8}$ − | 535.8$_{\pm69.4}$ − | 1.26$_{\pm0.12}$ − |
| SNLI | 69.8$_{\pm0.6}$ + | 454.0$_{\pm77.0}$ − | 1.11$_{\pm0.03}$ + |
| MNLI | 57.3$_{\pm0.5}$ − | 465.6$_{\pm98.3}$ − | 1.09$_{\pm0.02}$ + |

To investigate the impact of objective-guided operators on `InstOptima`, we conducted ablative experiments to assess the performance of `InstOptima-N`, which eliminates the objective guidance in the operators. The experimental results on `FlanT5-small` are presented in Table 2. Based on the results in Table 1 and Table 2, it is evident that `InstOptima-N` achieves inferior objective values on most datasets, particularly in terms of ACCURACY and LENGTH. However, for the SNLI dataset, `InstOptima-N` obtains better results in ACCURACY and PERPLEXITY compared to `InstOptima`. These findings demonstrate the effectiveness of objective-guided operators. Nonetheless, the concept of objective-guided operators is still in its early stages and warrants further investigation in future studies.

In conclusion, the experimental results indicate that objective-guided operators obtain better performance across various datasets.

**RQ2: Does the number of evolution generations matter in `InstOptima`?**

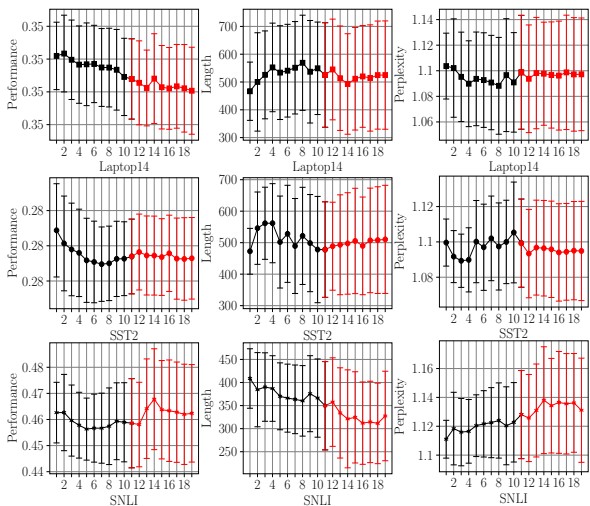

Figure 2: The trajectory plots of objective values across different datasets. We plot the trajectories of 10 additional generations using red lines. In these figures, lower objective values indicate better performance.

Generally, a larger number of generations tends to result in better objective values after optimization. We conducted additional training for 10 generations on the `Laptop14`, `SST2`, and `SNLI` datasets to study the significance of number of generations. Based on the experimental results in Fig. 2., in most cases (e.g., `Laptop14` and `SNLI` datasets), we observed a significant trade-off among the three objectives. However, due to the small scale of the evaluation data and population size, there were large variances in the performance objective (see the left column in Fig. 2). These variances in performance interfere with the convergence of the other two objectives, resulting in the absence of clear descending trends for the length and perplexity objectives with an increase in generations. However, this issue can be addressed by increasing the population size, number of generations, and scale of training data.

In conclusion, given the limited evaluation resources, the number of evolution generations showed limited improvement. Instead, it is important to reconcile different objective values to achieve the final instruction population.

**RQ3: Are there trade-offs between different objectives?**

To analyze the relationship between different objectives, we plot the Pareto front (refer to Fig. 5) of instructions into three groups. The two-dimensional

Pareto fronts between pairwise objectives are presented in Fig. 3.

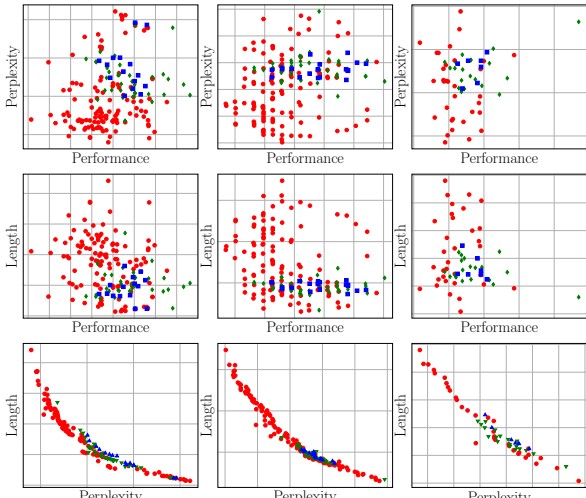

Figure 3: Visualizations of the 2D-Pareto fronts searched by `InstOptima` on three datasets. The three columns from left to right indicate the results on `Laptop14`, `SST2` and `SNLI` datasets, respectively.

Overall, there is a clear trade-off between instruction length and perplexity. However, when considering the pairs of performance-length and performance-perplexity, there is no clear trade-off observed in Fig. 3. This could be attributed to the lack of strict trade-offs and the presence of noise fitness points due to the evaluation of metrics on small datasets during optimization. It is expected that this issue can be mitigated when evaluating performance on larger datasets.

Nevertheless, `InstOptima` consistently discovers high-quality instructions in most scenarios, regardless of the loose trade-offs between objective pairs such as performance-length and performance-perplexity. This demonstrates the effectiveness of `InstOptima` in obtaining a diverse set of instructions.

## 4 Conclusion

We propose a multi-objective instruction optimization framework to obtain a diversified set of instructions. To address the challenges posed by the large and non-differentiable text search space, `InstOptima` utilizes objective-guided instruction operators based on LLM, which shows impressive performance in instruction generation. However, it is important to note that multi-objective instruction optimization is still in the early stages and requires further research in the future.

## 5 Limitations

The first limitation of `InstOptima` lies in the potential crisis of local optima in the multi-objective optimization. `InstOptima` initializes the instruction population based on fixed manually crafted instructions, which are then mutated using LLM. Although `InstOptima` has been demonstrated to search for diversified and high-quality instructions in experiments, the essence on fixed initial instructions may lead to traps in local optima during the multi-objective process. In the future, the generation of initial instruction populations, such as employing randomized initial instructions, remains a topic worth exploring.

The second limitation of `InstOptima` is related to experimental resources. Due to resource constraints, we only utilized single-round API calls to generate new instructions using LLM. This approach overlooks the contextual information that could help in understanding objective feedback in the instruction generation. We believe that continuous dialogue with LLM will significantly improve the quality of instruction generated by LLM. Additionally, due to the difficulty of accessing LLM, we conducted experiments with smaller population sizes and fewer iterations, which may underestimate the performance of `InstOptima`.

## Acknowledgments

This work was supported in part by the UKRI Future Leaders Fellowship under Grant MR/S017062/1 and MR/X011135/1; in part by NSFC under Grant 62376056 and 62076056; in part by the Royal Society under Grant IES/R2/212077; in part by the EPSRC under Grant 2404317; in part by the Kan Tong Po Fellowship (KTP\R1\231017); and in part by the Amazon Research Award and Alan Turing Fellowship.

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

# A  Appendix

## A.1  Experiment Setup

### A.1.1  Datasets

We selected six datasets for three classification tasks. For the aspect-based sentiment analysis (ABSA) task, we used the `Laptop14` and `Restaurant14` datasets (Pontiki et al., 2014). For text classification (TC) tasks, we chose the `SST2` (Socher et al., 2013) and `AGNews` (Zhang et al., 2015) datasets. We selected the `SNLI` (Bowman et al., 2015) and `MNLI` (Wang et al., 2019) datasets for the natural language inference (NLI) task. We trained our models on the first 1000 samples from the original training, validation and testing datasets, respectively.

### A.1.2  Experimental PLMs

For the LLM to operate instructions, we select the `ChatGPT`[4](OpenAI, 2023) with a temperature of 1 and a maximum token length of 500.

To obtain the objective value of performance, we performed instruction-based classification experiments using the `FlanT5-small` and `FlanT5-base` models (Chung et al., 2022), as well as `ChatGPT`, which are the latest and popular PLM/LLM for instruction learning. For the calculation of semantic complexity, we employed the `RoBERTa` (Liu et al., 2019) model from transformers(Wolf et al., 2020).

### A.1.3  Hyper-parameter Settings

The generation size and number of generation for `NSGA-II` is 100 and 10, respectively. In the fine-tuning[5] of the PLMs (i.e., `FlanT5-small` and `FlanT5-base`), we set the learning rate and batch size to $5e-5$ and 16, respectively. We fine-tune the PLMs for 3 epochs with an $L_2$ regularization parameter of 0.01.

### A.1.4  Experimental Environment

The experiments are carried out on a computer running the Cent OS 7 operating system, equipped with an RTX 3090 GPU and a Core i-12900$k$ processor. We use the `PyTorch` 2.0.0 library and `transformers` 4.28.0.

---

[4]`ChatGPT-turbo-0301` version.
[5]We use the Huggingface Trainer for fine-tuning, and the code is available in the supplementary materials.

## A.2  Additional Experiments for Summarization

### A.2.1  Generative Text Summarization

We conducted experiments for a text generation task. i.e., generative summarization. To evaluate `InstOptima`, we used three subsets from The `GigaWord` dataset and the `FlanT5-small` model in our experiments. In these subsets, the training set contains 5k training examples, while the testing set and validation set each have 1k examples. According to the Rouge$_1$ metric, it is evident that `InstOptima` performs well on the `GigaWord` dataset, demonstrating that it is a task-agnostic method for multi-objective instruction optimization.

### A.2.2  Experiments based on Different Backbone Models

We have conducted experiments to demonstrate the relationship between the backbone model and performance. Due to resource limitations, we are currently using `FlanT5` variants (small, base, and large, Llama is not implemented currently) as backbones to implement `InstOptima`. We have generated a box plot to visualize the experimental results in Fig. 4 The figure illustrates that performance is highly dependent on the scale of the backbone instruction-follow model. In other words, because the `FlanT5-small` model has limited capability to follow instructions, the accuracy achieved by an instruction is low and exhibits a larger variance compared to the larger instruction-follow models. In this context, `InstOptima` plays a crucial role in identifying instructions with optimized objectives.

## A.3  The Visualization of Pareto-fronts

In Fig. 5, we show the visualizations of Pareto-front instructions obtained by `InstOptima` on the `Laptop14`, `SST2` and `SNLI` datasets. Due to resource limitations, we only present the plots on the `Laptop14`, `SST2`, and `SNLI` datasets. We plot the first three fronts searched by `NSGA-II`, and the first three fronts are indicated by red, green, and blue colors, respectively.

## A.4  Multi-objective Optimization Algorithm

`InstOptima` is a multi-objective instruction optimization approach that evolves a population of instructions through a series of steps. We present the pseudo-code of `InstOptima` in Algorithm 1.

Table 3: The experimental performance of InstOptima. We show the ACCURACY instead of the performance objective for intuitive evaluation. The symbols ↗ and ↘ indicate larger is better and lower is better, respectively. We repeat each experiment in five rounds and report the average results. The best results are in **bold**. The ACCURACY is the best accuracy in the Pareto-front, while the LENGTH and PERPLEXITY are correlated with the instruction that achieves the best accuracy.

| MODEL | DATASET | InstOptima | | | RanInstruct | | | NoInstruct |
|---|---|---|---|---|---|---|---|---|
| | | ACCURACY ↗ | LENGTH ↘ | PERPLEXITY ↘ | ACCURACY ↗ | LENGTH ↘ | PERPLEXITY ↘ | ACCURACY ↗ |
| FlanT5-small | GigaWord | **33.7**$_{\pm 0.3}$ | **586.9**$_{\pm 91.5}$ | **1.08**$_{\pm 0.02}$ | 32.9$_{\pm 1.9}$ | 891.6$_{\pm 151.5}$ | **1.11**$_{\pm 0.03}$ | 30.8$_{\pm 0.8}$ |

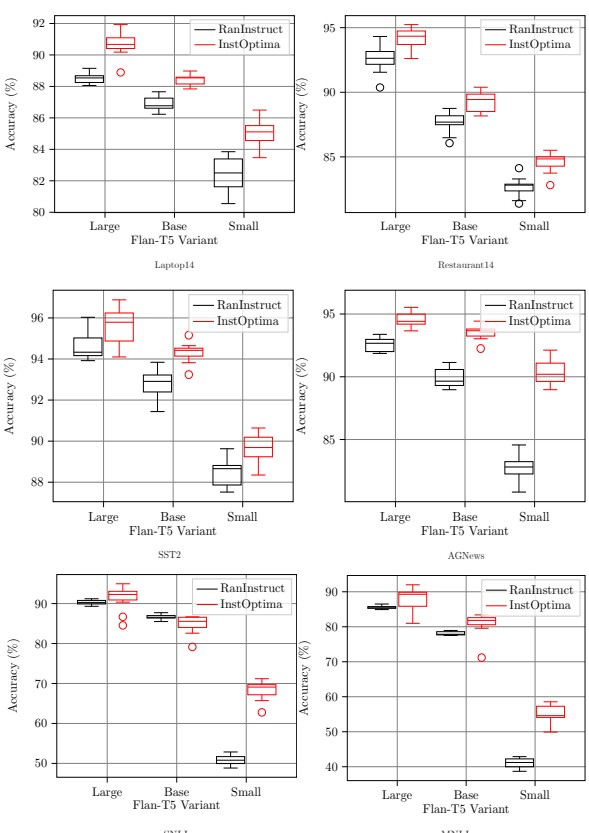

Figure 4: Box plot visualizations of the performance based on different backbone models.

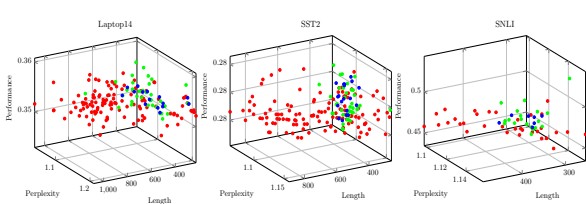

Figure 5: Visualizations of the Pareto fronts searched by InstOptima on three datasets. The PLM used to evaluate performance is FlanT5-small.

Firstly, the algorithm initializes a population of instructions. Then, it iteratively performs the following steps for a specified number of generations: selecting two instructions from the population, evaluating their objectives, applying LLM-based instruction operators to create new instructions, and adding them to a temporary population. After each generation, the temporary population is combined with the original population, and a selection process is applied to choose the fittest instructions. Finally, the algorithm returns the evolved population of instructions as the final results.

## A.5 Fixed Prompts for Instruction Operators

The prompts in green are the trigger of objective-guided instruction generation.

**Algorithm 1:** The pseudo code of `InstOptima`.

**Input:** Task dataset $\mathcal{D}$, Number of generations $N$, Population size $M$, Instruction Operators $\tilde{\mathbf{P}}$
**Output:** Evolved population of instructions $\mathcal{P}^*$

1   $\mathcal{P} \leftarrow \text{InitializePopulation}(M)$ ;             `// Initialize the population`
2   **for** $i \leftarrow 1$ **to** $N$ **do**
3     $\mathcal{Q} \leftarrow \emptyset$ ;                   `// Initialize the offspring population`
4     **for** $j \leftarrow 1$ **to** $M$ **do**
5       $\mathbf{I}_1 \leftarrow \mathcal{P}_j$ ;                 `// Select parent instruction`
6       $\mathbf{I}_2 \leftarrow \text{random}(\mathcal{P})$ ;           `// Select random parent instruction`
7       $\mathcal{F}_1 \leftarrow \text{EvaluateObjectives}(\mathbf{I}_1)^6$ ;     `// Evaluate objectives for parent 1`
8       $\mathcal{F}_2 \leftarrow \text{EvaluateObjectives}(\mathbf{I}_2)$ ;      `// Evaluate objectives for parent 2`
9       $(\mathbf{d}_1, \mathbf{e}_1) \leftarrow \mathbf{I}_1$ ;       `// Extract definition and example from parent 1`
10      $(\mathbf{d}_2, \mathbf{e}_2) \leftarrow \mathbf{I}_2$ ;      `// Extract definition and example from parent 2`
11      $\mathcal{O} \leftarrow \text{random}(\tilde{\mathbf{P}})$ ;            `// Select a random operator`
12      **if** $\mathcal{O} == \tilde{P}_{dm}$ **then**
13        $\hat{\mathbf{d}}_{dm} \leftarrow \text{ChatGPT}(\text{Concat}(\tilde{P}_{dm}, \mathbf{d}_1, \mathcal{F}_1))$ ;     `// Generate mutated definition`
14        $\hat{\mathbf{I}} \leftarrow \text{Concat}(\hat{\mathbf{d}}_{dm}, \mathbf{e}_1)$ ;     `// Combine mutated definition with example`
15      **if** $\mathcal{O} == \tilde{P}_{dc}$ **then**
16        $\hat{\mathbf{d}}_{dc} \leftarrow \text{ChatGPT}(\text{Concat}(\tilde{P}_{dc}, \mathbf{d}_1, \mathcal{F}_1, \mathbf{d}_2, \mathcal{F}_2))$ ;    `// Generate crossoverd definition`
17        $\hat{\mathbf{I}} \leftarrow \text{Concat}(\hat{\mathbf{d}}_{dc}, \mathbf{e}_1)$ ;     `// Combine crossoverd definition with example`
18      **if** $\mathcal{O} == \tilde{P}_{em}$ **then**
19        $\hat{\mathbf{e}}_{em} \leftarrow \text{ChatGPT}(\text{Concat}(\tilde{P}_{em}, \mathbf{e}_1, \mathcal{F}_1))$ ;      `// Generate mutated example`
20        $\hat{\mathbf{I}} \leftarrow \text{Concat}(\mathbf{d}_1, \hat{\mathbf{e}}_{em})$ ;    `// Combine original definition with mutated example`
21      **if** $\mathcal{O} == \tilde{P}_{ec}$ **then**
22        $\hat{\mathbf{e}}_{ec} \leftarrow \text{ChatGPT}(\text{Concat}(\tilde{P}_{ec}, \mathbf{e}_1, \mathcal{F}_1, \mathbf{e}_2, \mathcal{F}_2))$ ;     `// Generate crossoverd example`
23        $\hat{\mathbf{I}} \leftarrow \text{Concat}(\mathbf{d}_1, \hat{\mathbf{e}}_{ec})$ ; `// Combine original definition with crossoverd example`
24      $\mathcal{Q} \leftarrow \mathcal{Q} \cup \{\hat{\mathbf{I}}\}$ ;             `// Add offspring to the population`
25     $\mathcal{Q}^* \leftarrow \text{CombinePopulations}(\mathcal{P}, \mathcal{Q})$ ;    `// Combine parent and offspring populations`
26     $\mathcal{P} \leftarrow \text{SelectPopulation}(\mathcal{Q}^*, M)$ ;     `// Select the best individuals for the next generation`
27   $\mathcal{P}^* = \mathcal{P}$ ;            `// Set the evolved population as the final population`
28   **return** $\mathcal{P}^*$ ;                   `// Return the evolved population`

Table 4: The fixed prompts used to implement LLM-based instructions. "**<Input>**" indicates the input of the operators. The green keywords are the triggers of objective-guided instruction generation.

| OPERATORS | PROMPTS | INPUT |
|---|---|---|
| $\tilde{P}_{dm}$ | I want you to be a professional prompt engineer. Now I am working on the multi-objective evolutionary prompt optimization, and I need your help to design and optimize the template prompt. Here I give you an example template prompt, please understand the meaning of the prompt and modify it. Given the minimization objectives, please be creative and output the paraphrased or mutated prompt. Please remove Minimization objectives in the output: **<Input>** | $(\mathbf{d}, \mathcal{F})$ |
| $\tilde{P}_{dc}$ | I want you to be a professional prompt engineer. Now I am working on the multi-objective evolutionary prompt optimization for sentiment analysis, and I need your help to design and optimize the template prompt. Here I give you two template prompts, please understand the meaning of the two prompts and crossover them into a new prompt. Given the minimization objectives, please be creative and output the generated new prompt based on the two examples. Please remove Minimization objectives in the output: **<Input>** | $(\mathbf{d}_1, \mathcal{F}_1, \mathbf{d}_2, \mathcal{F}_2)$ |
| $\tilde{P}_{em}$ | I want you to be a professional prompt engineer. Now I am working on the multi-objective evolutionary prompt optimization for sentiment analysis, and I need your help to design and optimize the template prompt. Here I give you two groups of examples for completing the prompt, please generate new examples to substitute the following examples and there are no more than two examples in the new prompt. Given the minimization objectives, please be creative and output the generated example in the same format. Please remove Minimization objectives in the output: **<Input>** | $(\mathbf{e}, \mathcal{F})$ |
| $\tilde{P}_{ec}$ | I want you to be a professional prompt engineer. Now I am working on the multi-objective evolutionary prompt optimization for sentiment analysis, and I need your help to design and optimize the template prompt. Here I give you two groups of examples for completing the prompt, please read the examples of the two groups of examples and crossover the examples into a new example group and there are no more than two examples in the new examples. Given the minimization objectives, please be creative and output the crossovered the examples. Please remove Minimization objectives in the output: **<Input>** | $(\mathbf{e}_1, \mathcal{F}_1, \mathbf{e}_2, \mathcal{F}_2)$ |