# OpenReview forum: "InstOptima: Evolutionary Multi-objective Instruction Optimization via Large Language Model-based Instruction Operators"
_EMNLP/2023/Conference — EMNLP 2023 Findings_

### Official Review · Reviewer_YhoU · 2023-08-05

**Typos Grammar Style And Presentation Improvements:** Caption of Table 1 is strange (the "We")
**Soundness:** 3

**Excitement:**

3: Ambivalent: It has merits (e.g., it reports state-of-the-art results, the idea is nice), but there are key weaknesses (e.g., it describes incremental work), and it can significantly benefit from another round of revision. However, I won't object to accepting it if my co-reviewers champion it.

**Missing References:**

Figure 1 can add some elements to indicate the starting point of the iteration


**Paper Topic And Main Contributions:**

The paper proposed InstOptima, a multi-objective framework for achieving a varied set of improved instructions. It relies on LLMs like chatgpt to optimize the instructions (experimented with text classification tasks) on several pre-defined dimensions, including Performance, Length and Perplexity. It involves an iterative process. Experiments are done on six datasets for three classification tasks (by finetuning FlanT5-small and FlanT5-base with instructions as classification models).  In general, InstOptima achieves superior objectives in most cases

Overall I think this paper can have some value in LLM era, but there are some concerns.

**Questions For The Authors:**

1. What is "Crowdness Evaluation" in details?
2. What is the detail of RanInstruct? What is the prompt used here?
3. Why we want "Lower perplexity" of instructs? The author claimed "indicates that instructions are more easily understood by language models." But I am not sure what does this imply, a better generalization?

**Reasons To Accept:**

1. A clear method and framework for improving instructions with the help of LLMs.
2. Optimizing multiple pre-defined dimensions is a promising and reasonable idea.

**Reasons To Reject:**

1. Although this is a short paper, there are still many details that need to be clarified to make this paper self-contained.
2. Experimented classification model is T5 instead of an LLM, does this mean this method only applied to medium size models? I think this is acceptable but the authors should explain this.
3. The writing and figures can be improved

**Reproducibility:**

3: Could reproduce the results with some difficulty. The settings of parameters are underspecified or subjectively determined; the training/evaluation data are not widely available.

**Reviewer Confidence:**

4: Quite sure. I tried to check the important points carefully. It's unlikely, though conceivable, that I missed something that should affect my ratings.

---

> ### Author Rebuttal · Authors · 2023-08-28
>
> **R1**: We are checking the soundness of this paper, and will actively work on refining the clarity and self-containment of the paper. More specifically, we will rearrange the contents in the main sections and appendix section, and we will add details and references to the method section to make it self-contained.
>
> **R2**: We provide the experimental results (on two datasets currently) of InstOptima using ChatGPT as the backbone model (no fine-tuning) in the following tables:
>
> | InstOptima (ChatGPT) |  |  |  |
> | --- | --- | --- | --- |
> | Dataset | Accuracy $\nearrow$ | Length $\searrow$ | Perplexity $\searrow$ |
> | Laptop14 | 83.2$_{\pm 2.2}$ | 512.9$_{\pm 51.5}$ | 1.08$_{\pm 0.02}$ |
> | Restaurant14 | 96.3$_{\pm 1.9}$ | 487.3$_{\pm 55.9}$ | 1.09$_{\pm 0.02}$ |
>
> | RanInstruct |  |  |  |
> | --- | --- | --- | --- |
> | Dataset | Accuracy $\nearrow$ | Length $\searrow$ | Perplexity $\searrow$ |
> | Laptop14 | 83.1$_{\pm 0.8}$ | 877.6$_{\pm 51.5}$ | 1.05$_{\pm 0.03}$ |
> | Restaurant14 | 92.1$_{\pm 1.3}$ | 421.6$_{\pm 82.4}$ | 1.10$_{\pm 0.02}$ |
>
> | NoInstruct |  |
> | --- | --- |
> | Dataset | Accuracy $\nearrow$ |
> | Laptop14 | 67.8$_{\pm 5.8}$ |
> | Restaurant14 | 75.2$_{\pm 6.1}$ |
>
> where the experimental results show the effectiveness of InstOptima for ChatGPT. Due to the tight response period, the experiments are evaluated on two datasets, but we will share more experimental results on the rest of the three datasets in the appendix.
>
> **R3**:  We are proofreading the manuscript to improve its writing, and we will rework Figure 1 to show evolution details.
>
> **Q1**: Crowding distance is a concept used in multi-objective optimization to assess the diversity and distribution of solutions within a Pareto front. The calculation aims to provide a measure of how crowded the solutions are in the objective space, which helps in maintaining a diverse set of solutions. In other words, the crowding distance calculation measures how much an individual is "crowded" by comparing the distances between the neighbors along each objective dimension. Larger differences in objective values contribute to larger crowding distances, indicating that an individual is more spaced out from its neighbors. This calculation helps in selecting a diverse set of solutions for further stages of the optimization process. Please refer to the supplementary materials for the code implementation, i.e., lines 272 - 292 in nsga2.py.
>
> **Q2**: We employ ChatGPT to generate a population of instructions using hand-crafted seed instructions. This process is similar to InstOptima, yet without using an evolution process. The instructions and performance results obtained through InstOptima have already been made available in the supplementary materials. Additionally, we will include the random instructions and their corresponding results in the revised version.
>
> **Q3**: Yes, it is useful for generalization. The reason why we need low perplexity is because:
>
>     (1) A higher perplexity indicates that the words in the text are not well-organized. For example, text edits based on word deletions and insertions can lead to grammatical or syntactical errors, resulting in higher perplexity compared to natural texts.
>
>     (2) The perplexity issue is particularly significant in the language generation of small language models. Therefore, generating instructions that are easy to read and that reduce the risk of misinterpretation is important for both language models and instruction engineers.
>
> Thank you for your questions. We hope that your concerns have been addressed. All of the responses will be included in the revision. We are looking forward to answering other questions in the discussion period.

---

### Official Review · Reviewer_RjTW · 2023-08-05

**Soundness:** 3

**Excitement:**

3: Ambivalent: It has merits (e.g., it reports state-of-the-art results, the idea is nice), but there are key weaknesses (e.g., it describes incremental work), and it can significantly benefit from another round of revision. However, I won't object to accepting it if my co-reviewers champion it.

**Paper Topic And Main Contributions:**

This paper presents a new method called InstOptima, which treats instruction generation as a multi-objective optimization problem. This method uses a large language model (LLM) to simulate the generation of instruction operations using a genetic algorithm, including mutation and crossover. Meanwhile, an objective-guided mechanism is introduced to help the LLM generate high-quality instructions. InstOptima divides the direction of instruction search into multiple objectives to achieve fine control over instruction quality. Using multi-objective optimization algorithms helps to automatically search for a set of high-quality instructions and improve the performance.

**Questions For The Authors:**

1. What was the initial motivation for using a genetic algorithm?
2. How is initialization implemented?
3. The number of evolutionary iterations in A.2 does not seem to affect perplexity, and the fluctuation of perplexity is large. The paper claims that this can be solved by increasing the population size, number of generations, and scale of training data. Can you provide some quantified results to support this claim?
4. Can we replace the random operator part with traditional text augmentation methods (such as synonym replacement, back translation, typos, etc.)?
5. What is the extra search time introduced by InstOptima compared to random instruction?


**Reasons To Accept:**

1. Compared to the random instruction generation method of RanInstruct, the comprehensive multi-objective optimization considering instruction length and perplexity is more reasonable.
2. Good performance, which provides significant improvement over the random RanInstruct.


**Reasons To Reject:**

While the contribution claims that the proposed method enhances robustness against adversarial instruction attacks, the current experiments do not provide sufficient evidence to substantiate this claim. Although relevant findings from prior studies are cited, it is important to experimentally verify their applicability to the specific scenario addressed in this paper. Additional rigorous experiments are needed to demonstrate the efficacy of the proposed instructions in defending against such attacks.

The comparison of algorithms conducted by the paper is somewhat limited, considering the constraints of this short paper. It would be beneficial to include a broader range of algorithms for comparison to ensure a more comprehensive evaluation and a clearer understanding of the proposed method's performance. Some quantified results would facilitate a more robust analysis and reinforce the significance of the work.

**Reproducibility:**

4: Could mostly reproduce the results, but there may be some variation because of sample variance or minor variations in their interpretation of the protocol or method.

**Reviewer Confidence:**

3: Pretty sure, but there's a chance I missed something. Although I have a good feel for this area in general, I did not carefully check the paper's details, e.g., the math, experimental design, or novelty.

---

> ### Author Rebuttal · Authors · 2023-08-28
>
> **R1**: Due to resource and time limitations, organizing the specific experiments related to adversarial instruction attacks cannot be finished in the rebuttal period. However, We have decided to incorporate these experiments in the appendix to ensure comprehensive coverage. Please kindly allow us some time to sort out the robustness evaluation details.
>
> **R2**: We will incorporate comparisons towards state-of-the-art instruction generation methods. As a complementary experiment, we have reported new results of InstOptima based on the ChatGPT backbone:
>
> | InstOptima |  |  |  |
> | --- | --- | --- | --- |
> | Dataset | Accuracy $\nearrow$ | Length $\searrow$ | Perplexity $\searrow$ |
> | Laptop14 | 83.2$_{\pm 2.2}$ | 512.9$_{\pm 51.5}$ | 1.08$_{\pm 0.02}$ |
> | Restaurant14 | 96.3$_{\pm 1.9}$ | 487.3$_{\pm 55.9}$ | 1.09$_{\pm 0.02}$ |
>
> | RanInstruct |  |  |  |
> | --- | --- | --- | --- |
> | Dataset | Accuracy $\nearrow$ | Length $\searrow$ | Perplexity $\searrow$ |
> | Laptop14 | 83.1$_{\pm 0.8}$ | 877.6$_{\pm 51.5}$ | 1.05$_{\pm 0.03}$ |
> | Restaurant14 | 92.1$_{\pm 1.3}$ | 421.6$_{\pm 82.4}$ | 1.10$_{\pm 0.02}$ |
>
> | NoInstruct |  |
> | --- | --- |
> | Dataset | Accuracy $\nearrow$ |
> | Laptop14 | 67.8$_{\pm 5.8}$ |
> | Restaurant14 | 75.2$_{\pm 6.1}$ |
>
> **Q1**: First, it is because the genetic algorithm is an easy-to-implement method for multi-objective instruction optimization. Therefore, the experiments are easy to reproduce. Everyone can change the evolution engine on their demands. Second, the text edition-based operators of genetic algorithms have been proven to be effective for text evolution [1][2]. This inspires our LLM-based operators.
>
> **Q2**: We first define the instruction GA operators based on ChatGPT. We hand-crafted ten seed instructions (We will clarify these seed instructions in the revision.) and used GA operators to manipulate the seed instructions. The new instructions operated by ChatGPT are combined into a population. We will update Figure 1 to indicate the start of initialization.
>
> **Q3**: Figure 2 visualized the trajectories of a single individual with the best accuracy, so it cannot properly visualize the evolution progress. We have fixed this problem by visualizing the average accuracy, length and perplexity within the population and prepared the fixed figure on the Laptop14 dataset. In the fixed figure, we do not need to improve the population size, number of generations, and scale of training data to show clear convergence trajectories. Please refer to https://anonymous.4open.science/status/InstOptima-7C28 to find the population’s convergence trajectories. In the new figure, please note that the instruction lengths are growing gradually because our seed instructions in initialization are too short compared to Pareto front instructions.
>
> **Q4**: No, this will lead to poor diversity as well as poor accuracy of instructions in our proof-of-concept experiments.
>
> **Q5**: We report the average time costs in a population. Please refer to Q4 in the rebuttal of Reviewer WrSn. The search time of InstOptima is approximately calculated by multiplying # of generations, # of individuals in a population, and the evaluation budget for each individual, while the random instruction requires no extra search time budget.
>
> Finally, thank you for your careful review. We hope that your concerns have been addressed. All of the responses will be included in the revision. We are looking forward to answering other questions in the discussion period.
>
> [1] Alzantot, Moustafa, et al. "Generating natural language adversarial examples." *arXiv preprint arXiv:1804.07998* (2018).
>
> [2] Zang, Yuan, et al. "Word-level textual adversarial attacking as combinatorial optimization." *arXiv preprint arXiv:1910.12196* (2019).

---

### Official Review · Reviewer_WrSn · 2023-08-09

**Typos Grammar Style And Presentation Improvements:** 535
**Soundness:** 4

**Excitement:**

3: Ambivalent: It has merits (e.g., it reports state-of-the-art results, the idea is nice), but there are key weaknesses (e.g., it describes incremental work), and it can significantly benefit from another round of revision. However, I won't object to accepting it if my co-reviewers champion it.

**Paper Topic And Main Contributions:**

The authors investigate the problem of generating instructions for in-context learning and define objectives such as performance, length, and perplexity that affect the quality of the instructions. They propose InstOptima, which treats instruction generation as an evolutionary multi-objective optimization problem. InstOptima randomly selects an operator (definition formulation/definition crossover/example mutation/example crossover) to evolve the instruction by feeding the objectives back. The authors report that InstOptima achieves better performance on various datasets.

**Questions For The Authors:**

Q1: What is the performance of ChatGPT on the five datasets?
Q2: Why not just use F1 as performance objective in Section 2.2.2?
Q3: Is the EvaluateObjectives() function in algorithm 1 evaluated on the validation set? More details need to be provided.
Q4: What are the time costs of the entire algorithm? It seems that multiple evaluations are conducted during generation.

**Reasons To Accept:**

A1: The proposed method is novel and makes sense.

A2: The paper is easy to read and comprehend.

A3: The experimental results demonstrate the effectiveness of the proposed method.

**Reasons To Reject:**

R1: The organization of the paper needs improvement. For instance, crucial sections like experimental settings and RQ2 and RQ3 in section 3.2 are only presented in the Appendix. Additionally, the paper lacks related work. Therefore, I suggest that the paper should be reformatted to a long paper.

R2: According to Table 1, the performance gap between InstOptima and RanInstruct is significantly influenced by the backbone model used. For instance, the gains are not substantial when FlanT5-base is used. Hence, it is better to employ more powerful backbones, such as llama.

R3: The paper fails to compare InstOptima with other instruction generation methods, like self-instruct.

**Reproducibility:**

4: Could mostly reproduce the results, but there may be some variation because of sample variance or minor variations in their interpretation of the protocol or method.

**Reviewer Confidence:**

4: Quite sure. I tried to check the important points carefully. It's unlikely, though conceivable, that I missed something that should affect my ratings.

---

> ### Author Rebuttal · Authors · 2023-08-28
>
> **R1**: This is a timely pilot work so we organized it as a short paper. We will move RQ2 and RQ3 to the Experiments section. Meanwhile, we will include a discussion section on related works. But we may not be able to reformat the current submission into a regular long paper, we are committed to enhancing the overall presentation and readability to the best of our abilities. For example, we will highlight the key contents with colored fonts to assist readers in navigating and focusing on their interested sections.
>
> **R2**: Upon further consideration of your concern, another question to ponder is how to unveil the degree of dependency between the backbone model and performance. We are introducing a new specific section that will present experimental results of other backbone models (e.g., Llama, Vicuna) to answer this question.
>
> **R3**: This is going to be addressed in the revision. We will incorporate performance comparisons towards multiple state-of-the-art instruction-based models and present discussions about these experiments. We provide the experimental results of ChatGPT on two datasets in the following table:
>
> | InstOptima |  |  |  |
> | --- | --- | --- | --- |
> | Dataset | Accuracy $\nearrow$ | Length $\searrow$ | Perplexity $\searrow$ |
> | Laptop14 | 83.2$_{\pm 2.2}$ | 512.9$_{\pm 51.5}$ | 1.08$_{\pm 0.02}$ |
> | Restaurant14 | 96.3$_{\pm 1.9}$ | 487.3$_{\pm 55.9}$ | 1.09$_{\pm 0.02}$ |
>
>
>
> **Q1**: Our experiments are conducted to prove it is effective for LLMs, including ChatGPT. We provide the performance of ChatGPT in the following tables:
>
> | RanInstruct |  |  |  |
> | --- | --- | --- | --- |
> | Dataset | Accuracy $\nearrow$ | Length $\searrow$ | Perplexity $\searrow$ |
> | Laptop14 | 83.1$_{\pm 0.8}$ | 877.6$_{\pm 51.5}$ | 1.05$_{\pm 0.03}$ |
> | Restaurant14 | 92.1$_{\pm 1.3}$ | 421.6$_{\pm 82.4}$ | 1.10$_{\pm 0.02}$ |
>
> | NoInstruct |  |
> | --- | --- |
> | Dataset | Accuracy $\nearrow$ |
> | Laptop14 | 67.8$_{\pm 5.8}$ |
> | Restaurant14 | 75.2$_{\pm 6.1}$ |
>
> where RanInstruct is ChatGPT based on the randomly generated instructions (i.e., definition and example), and NoInstruct means only examples (no definition) are provided. the experimental results show the effectiveness of InstOptima for ChatGPT. Due to the tight response period, the experiments are evaluated on two datasets, but we will share more experimental results on the rest of the three datasets in the appendix.
>
> **Q2**: When considering the generalizability of InstaOptima in other tasks, employing the combination of multiple metrics is a more scalable approach. To elaborate, when dealing with a composite of diverse metrics spanning various facets (which may occasionally exhibit conflicts), the evolutionary process can be effectively undertaken.
>
> **Q3**: It depends on the existence of a validation set, which means if there is no validation set, such as the Restaurant14 and Laptop14 datasets, the EvaluateObjectives evaluates the test set.
>
> **Q4**: We report the average time costs in a population and the time costs for FlanT5-small and FlanT5-base are shown in the following tables, respectively:
>
> | **Dataset** |**Performance (s)** | **Perplexity (s)**| **Length (s)** |
> |---|---|---|---|
> | Laptop14 | $138.72_{\pm 27.3}$ | $0.48_{\pm 0.01}$  |$6.24\times 10^{-5}$|
> |Restaurant14 | $149.82_{\pm 34.1}$  | $0.42_{\pm 0.02}$  |$6.88\times 10^{-5}$|
> | SST2| $117.76_{\pm 25.8}$ |$0.38_{\pm 0.01}$ |$7.55\times 10^{-5}$|
> | AGNews| $117.16_{\pm 23.5}$|$0.45_{\pm 0.02}$|$8.27\times 10^{-5}$|
> | SNLI | $400.46_{\pm 55.8}$| $0.39_{\pm 0.02}$   |$7.84\times 10^{-5}$|
> | MNLI | $879.17_{\pm 81.3}$|$0.44_{\pm 0.02}$  |$8.05\times 10^{-5}$|
>
> | **Dataset** |**Performance (s)** | **Perplexity (s)**| **Length (s)** |
> |---|---|---|---|
> | Laptop14 |$298.58_{\pm 55.7}$ | $0.38_{\pm 0.02}$  |$6.39\times 10^{-5}$|
> |Restaurant14 | $342.72_{\pm 49.7}$ |$0.35_{\pm 0.02}$| $7.81\times 10^{-5}$ |
> | SST2| $185.60_{\pm 32.1}$| $0.38_{\pm 0.02}$  |$6.53\times 10^{-5}$|
> | AGNews| $202.46_{\pm 38.1}$| $0.41_{\pm 0.02}$  |$8.81\times 10^{-5}$|
> | SNLI | $468.63_{\pm 48.8}$| $0.32_{\pm 0.02}$  |$7.56\times 10^{-5}$|
> | MNLI | $950.98_{\pm 77.8}$| $0.36_{\pm 0.01}$  |$7.14\times 10^{-5}$|
>
> Our experiments are based on an RTX3090 GPU. More details about the environment and implementation can be found in the appendix.
>
> Thank you for your insightful review. We hope that your concerns have been addressed. All of the responses will be included in the revision. We are looking forward to answering other questions in the discussion period.

---

### Meta-Review · Area_Chair_LZDo · 2023-09-24

**Recommendation:** 3

**Metareview:**

This paper presents a new method called InstOptima, which treats instruction generation as a multi-objective optimization problem. InstOptima randomly selects an operator (definition formulation/definition crossover/example mutation/example crossover) to evolve the instruction by feeding the objectives back. The reviewers did raise a few concerns and during the rebuttal period, the authors provided further explanations and experimental results, which solved the concerns to some extent. A critical though non-technical issue is that once accepted, the authors need to carefully reorganize the content into 5 pages so as to come up with a self-contained version.

---

### Decision · Program_Chairs · 2023-10-07

**Decision:**

Accept-Findings

**Comment:**

This paper presents a new method called InstOptima, which treats instruction generation as a multi-objective optimization problem. InstOptima randomly selects an operator (definition formulation/definition crossover/example mutation/example crossover) to evolve the instruction by feeding the objectives back. The reviewers did raise a few concerns and during the rebuttal period, the authors provided further explanations and experimental results, which solved the concerns to some extent. A critical though non-technical issue is that once accepted, the authors need to carefully reorganize the content into 5 pages so as to come up with a self-contained version.